# Short birth spacing and its association with maternal educational status, contraceptive use, and duration of breastfeeding in Ethiopia. A systematic review and meta-analysis

Yitayish Damtie[1]*, Bereket Kefale[1], Melaku Yalew[1], Mastewal Arefaynie[1‡], Bezawit Adane[2‡]

1 Department of Reproductive and Family Health, School of Public Health, College of Medicine and Health Sciences, Wollo University, Dessie, Ethiopia, 2 Department of Epidemiology and Biostatistics, School of Public Health, College of Medicine and Health Sciences, Wollo University, Dessie, Ethiopia

☯ These authors contributed equally to this work.
‡ These authors also contributed equally to this work.
* yitutile@gmail.com

## Abstract

### Background

Closely spaced birth increases the risk of adverse maternal and child health outcomes. In Ethiopia, the prevalence of short birth spacing was highly variable across studies. Besides, contraceptive use, educational status, and duration of breastfeeding were frequently mentioned factors affecting short birth spacing. Therefore, this meta-analysis aimed to estimate the pooled prevalence of short birth spacing and its association with contraceptive use, educational status, and duration of breastfeeding among reproductive-age women in Ethiopia.

### Methods

International databases: Google Scholar, PubMed, CINAHL, Cochrane library, HINARI, and Global Health were searched systematically to identify articles reporting the prevalence of short birth spacing and its association with contraceptive use, educational status, and duration of breastfeeding among reproductive-age women in Ethiopia. The data were analyzed by STATA/SE version-14 statistical software. The random-effect model was used to estimate the pooled prevalence of short birth spacing and the log odds ratio was used to determine the association. Moreover, egger's test and I-squared statistics were used to assess publication bias and heterogeneity respectively.

### Results

After reviewing 511 research articles, a total of nine articles with 5,682 study participants were included in this meta-analysis. The pooled prevalence of short birth spacing in Ethiopia was 46.9% [95% CI: (34.7, 59.1)]. Significant heterogeneity was observed between studies ($I^2$ = 98.4, $p$ <0.001). Not using contraceptives [OR = 3.87, 95% CI: (2.29, 6.53)] and

**Data Availability Statement:** All relevant data are within the manuscript and its Supporting Information files.

**Funding:** The authors received no specific funding for this work.

**Competing interests:** The authors have declared that no competing interests exist.

**Abbreviations:** EDHS, Ethiopia Demographic and Health Survey; HSTP, Health Sector Transformation Plan; NOS, Newcastle-Ottawa quality assessment Scale; PRISMA, Preferred Reporting Items for Systematic Reviews and Meta-Analysis; SSA, Sub-Saharan Africa; SDG, Sustainable Development Goals; WHO, World Health Organization.

duration of breastfeeding < 24 months [OR = 16.9, 95%CI: (2.69, 106.47)] had a significant association with short birth spacing.

## Conclusions

Although a minimum inter-pregnancy interval of two years was recommended by the World Health Organization (WHO), significant numbers of women still practiced short birth spacing in Ethiopia. Duration of breastfeeding and non-use of contraceptives were factors significantly associated with short birth spacing. So, efforts should be made to improve breastfeeding practice and contraceptive utilization among women in Ethiopia.

## Introduction

Maternal mortality remains the major public health challenge to the global population. Reports indicated that an estimated 303,000 maternal deaths occurred worldwide and among this, 99% of deaths were from developing countries [1]. The scenario is worst in sub-Saharan Africa (SSA) in which 546 deaths per 100,000 live births were documented as compared to 216 maternal deaths per 100,000 live births worldwide [2, 3]. The recent national health survey data showed that an estimated 412 maternal deaths per 100,000 live births occurred in Ethiopia [4].

Birth spacing, also known as the birth interval is the duration of time between two successive live births [5–7]. World Health Organization (WHO) recommends a minimum inter-pregnancy interval of two years or an inter-birth interval of thirty-three months or more to ensure the maximum health benefits for the mothers and the newborns [8, 9]. Spacing the child for a minimum of two years reduces infant mortality by 50% [10]. There is a significant variation in the practice of birth spacing across developing and developed countries [11]. Women from developing regions usually have short birth spacing than their actual preference [12]. Data from 52 developing countries indicated that over two-thirds of births happened within 30 months since the preceding live birth [13]. Like many other developing countries, short birth spacing is also the problem of Ethiopian in which 41.5% of women practiced short birth spacing [14].

Short birth spacing has been linked with different adverse pregnancy and childbirth outcomes such as low birth weight [15–18], preterm birth [15–20], congenital anomalies [21, 22], autism [23–27], small size for gestational age [15, 20], and neonatal, infant and child mortality [5, 15, 20, 28–31]. Moreover, women with short birth spacing are at high risk of developing hypertensive disorders of pregnancy, anemia, third-trimester bleeding, premature rupture of membranes, and puerperal endometritis [9, 10, 32, 33]. Beyond the maternal and child health implications, closely spaced birth increases population growth, decelerates one's country economic development, decreases women's productivity, and increases the demand for natural resources [9].

Reducing maternal and neonatal mortality is one of the key targets of Sustainable Development Goals (SDGs) particularly SDG 3 which aimed to reduce the global Maternal Mortality Ratio (MMR) to less than 70 per 100,000 live births and to decrease neonatal mortality below 12 per 1000 live births by the year 2030 [34]. The Ethiopian government is also implementing Health Sector Transformation Plan (HSTP) IV which aimed to reduce MMR from 420 to 199 per 100,000 live births and under five-year, infant and neonatal mortality rates from 64, 44 and 28 to 30, 20 and 10 per 1,000 live births respectively [35]. Optimal birth spacing is an important concept largely used for maternal and child health advocacy, designing family planning policies, and monitoring and evaluation of policies, strategies, and programs related to maternal and child health.

In Ethiopia, different studies were conducted on childbirth spacing [36–44] and a wide range of maternal and service-related factors like maternal educational status [39, 42, 43], contraceptive use [38, 39, 41–44], duration of breastfeeding [39, 42, 43] residence [38, 41], sex of the index child [39, 42], survival status of the index child [38], religion [40], and unwanted pregnancy [40] were identified. However, the prevalence of short birth spacing varies from region to region ranging from 23.3% to 59.9% [39, 40]. In addition to this, there is no country-level study assessing short birth spacing and associated factors among women of childbearing age in Ethiopia. Moreover, maternal educational status, contraceptive use, and duration of breastfeeding are frequently mentioned and clinically important factors affecting short birth spacing even though they have controversial findings across the included articles. Therefore, this meta-analysis aimed to estimate the pooled prevalence of short birth spacing and its association with maternal educational status, contraceptive use, and duration of breastfeeding among reproductive-age women in Ethiopia.

The result of this study will serve as an input for program designers and policymakers to design evidence-based interventions related to child spacing. It will also have paramount importance for future researchers interested in related topics.

## Materials and methods

### Search strategy

A systematic search was conducted on Google Scholar, PubMed, CINAHL, Cochrane library, HINARI, and Global Health to find both published and unpublished research articles. Besides, Addis Ababa Digital Library was also searched to identify unpublished papers. Grey literatures were identified through the input of content experts and the review of reference lists. The searching was carried out from April 1 up to May 30, 2020, and articles published from 1990 up to May 30, 2020, were included in the review. The Endnote software was used for managing articles and removing duplicates identified by our search strategy. The search strategy includes the following keywords: "proportion", "magnitude", "prevalence", "incidence", "Birth spacing", "child spacing", "birth interval", "suboptimal birth intervals", "short birth spacing", "suboptimal child spacing", "optimal birth spacing", "inter-birth interval", "risk factors", "predictors", "factors", "determinants", "associated factors", "married women", "women", "women of childbearing age", "Ethiopia" independently and in combination using "OR" or "AND" Boolean operators (see S1 File). This systematic review and meta-analysis was organized according to the Preferred Reporting Items for Systematic Reviews and Meta-Analysis (PRISMA-2009) checklist [45] (see S1 Table). The protocol of this systematic review and meta-analysis was registered in the international prospective register of systematic reviews (PROSPERO) with a specific registration number: CRD42020160922.

### Inclusion and exclusion criteria

Studies conducted in Ethiopia, studies involving women of childbearing age, all types of observational studies (Cross-sectional, case-control, and cohort), published and unpublished articles, full-text articles, articles published from 1990 up to May 30, 2020, and articles written in the English language were included in this study. Whereas relevant articles with full texts of which unavailable after two email contacts of the corresponding authors were excluded.

### Outcome measurement

This review measured two main outcomes. The first outcome was the overall pooled prevalence of short birth spacing which was computed by dividing the number of women with short

birth spacing to the total sample size multiplied by 100. The second outcome was the association between the duration of breastfeeding, contraceptive use, maternal educational status, and short birth spacing.

## Data extraction

Three authors (YD, BK, and MY) independently extracted all the necessary data using a standardized data extraction format. The remaining two authors (MA and BA) solved the disagreement raised at the time of data extraction. The corresponding author of the research article was also contacted for clarification and additional information. Data extraction form includes author name, region, study area, publication year, study design, study setting, sample size, response rate, and the number of women with short birth spacing. For the associated factors, frequencies in the form of two by two tables were extracted, and the log odds ratio for each factor was calculated accordingly.

## Quality assessment

BK and MY independently assessed the quality of each research article using the Newcastle-Ottawa quality assessment scale (NOS) [46]. The mean score was taken to solve the disagreements between the two authors.

The quality assessment tool has three subdivisions. The first segment deals with the methodological quality, the second subdivision mainly focuses on the comparability of the study, and the third section deals with the statistical analysis and the outcome of the research article. Finally, studies scoring $\geq 6$ out of 10 scales were considered as high-quality research articles.

## Data analysis

The relevant data were extracted using a Microsoft Excel spreadsheet and exported into STATA/SE version-14 statistical software for analysis. The heterogeneity between the included articles was assessed by using a $p$-value for the $I^2$ test [47]. The random-effect model was used to estimate the Der Simonian and Laird's pooled effect as a result of significant heterogeneity between the studies. Besides, subgroup analysis by residence, sample size, and the cutoff point used to measure the outcome variable was done to decrease the random variations among the point estimates of original articles.

Univariate meta-regression analysis was also done by taking sample size, publication year, and response rate as covariates. Moreover, the presence of publication bias was assessed using both the funnel plot and Egger's test at a 5% significant level [48].

The point estimates with their 95% confidence interval were presented using the forest plot and the log odds ratio was used to determine the association between short birth spacing and maternal educational status, contraceptive use, and duration of breastfeeding.

## Results

### Study selection

A total of five hundred eleven (511) published and unpublished studies were identified through electronic databases (Google Scholar, PubMed, CINAHL, Cochrane library, Hinari, and Global Health) and digital library search. Of these identified articles, 502 of them were dropped as a result of duplication, by the titles and abstracts, due to the absence of full texts, and as a result of not fulfilling the eligibility criteria. Finally, 9 eligible articles were included for analysis (Fig 1).

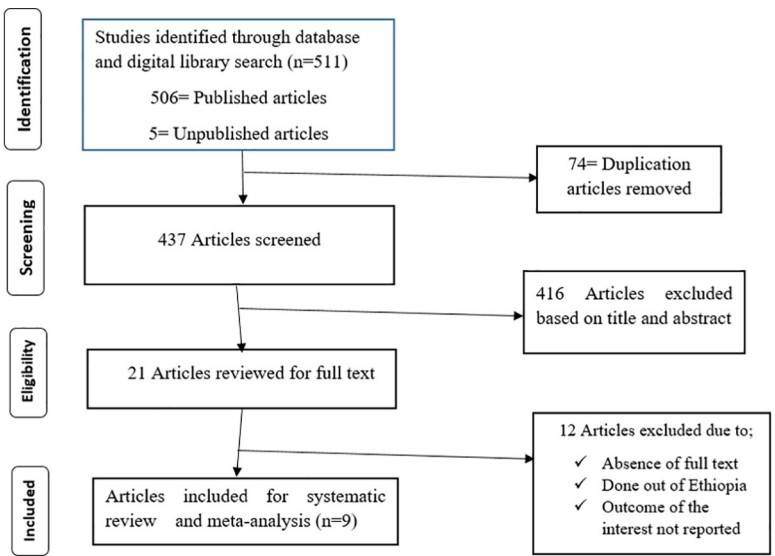

**Fig 1. PRISMA flow diagram describes the selection of studies for a systematic review and meta-analysis of short birth spacing and its association with maternal educational status, contraceptive use, and duration of breastfeeding in Ethiopia, 2020.**

## Characteristics of the included studies

A total of 9 research articles published from 2011 up to 2019 were included in this meta-analysis. Among the 9 articles, 6 of them were cross-sectional [36–41], 2 of them were case-control [42, 43] and the remaining 1 study was a retrospective follow-up study [44]. This meta-analysis included 4 studies from Oromia region [36, 38, 39, 43], 2 studies from Amhara region [37, 44], 2 studies from SNNP region [41, 42] and 1study from Tigray region [40] with a sample size ranged from 314 [39] to 811 [38, 41]. From the 9 articles, 6 studies with a total of 3,797 women of childbearing age were included to estimate the pooled prevalence of short birth spacing in Ethiopia. The lowest prevalence of short birth spacing (23.3%) was reported from a study done in the Tselemti district, Tigray region [40] whereas the highest prevalence of short birth spacing (59.9%) was observed from a study done in Serbo town, Oromia region [39]. (Table 1).

**Table 1. Descriptive summary of nine studies included in the meta-analysis of short birth spacing and associated factors among women of childbearing age in Ethiopia, 2020.**

| Authors | Publication year | Region | Study Area | Study Design | Sample size | Response rate | Prevalence (%) | Quality score |
|---|---|---|---|---|---|---|---|---|
| Shallo and Gobena [36] | 2019 | Oromia | Dodota woreda | cross-sectional | 647 | 98 | 49.1 | 7 |
| Ejigu et al. [37] | 2019 | Amhara | Debre Markos | cross-sectional | 411 | 98.3 | 40.9 | 6 |
| Tsegaye et al. [38] | 2017 | Oromia | Illubabor zone | cross-sectional | 811 | 98.2 | 51.2 | 8 |
| Ayane et al [39] | 2019 | Oromia | Serbo town | cross-sectional | 314 | 100 | 59.9 | 6 |
| Gebrehiwot et al [40] | 2019 | Tigray | Tselemti | cross-sectional | 803 | 99.6 | 23.3 | 7 |
| Yohannes et al [41] | 2011 | SNNP | Lemo district | cross-sectional | 811 | 96.1 | 57 | 8 |
| Hailu et al [42] | 2016 | SNNP | Arba Minch | Case-control | 636 | 100 | – | 7 |
| Begna et al [43] | 2013 | Oromia | Yaballo Woreda | Case-control | 636 | 97.5 | – | 8 |
| Tessema et al. [44] | 2013 | Amhara | Dabat district | Retrospective follow up | 613 | 99.5 | – | 6 |

SSNP; Southern Nations, Nationalities, and Peoples.

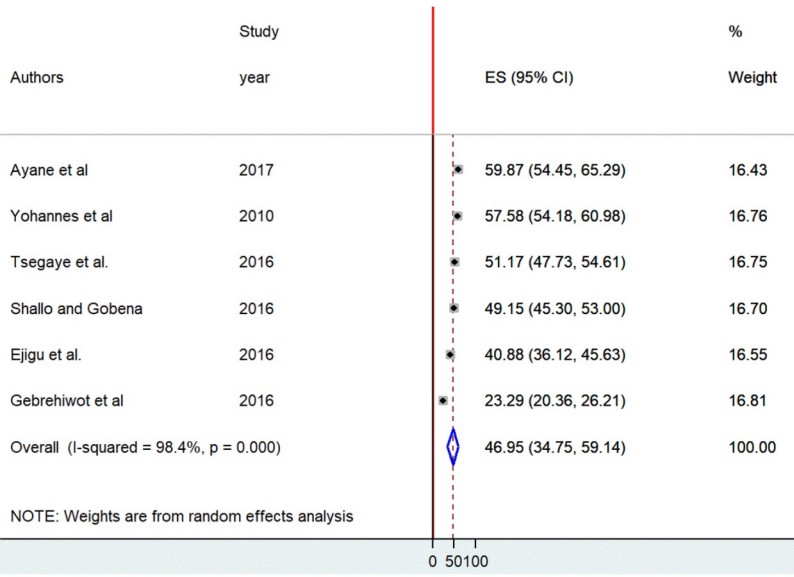

**Fig 2. Forest plot of the pooled prevalence of short birth spacing among reproductive-age women in Ethiopia, 2020.**

## Prevalence of short birth spacing in Ethiopia

The pooled prevalence of short birth spacing from the six included studies [36–41] in Ethiopia was 46.9%, [95% CI: (34.7, 59.1)]. A random-effect meta-analysis model was used to estimate the pooled prevalence as a result of substantial heterogeneity among the included studies ($I^2$ = 98.4, $p$ <0.001) (Fig 2).

To identify the possible source of heterogeneity, univariate meta-regression was conducted using sample size, publication year, and response rate as a factor, but neither of them were not a significant source of heterogeneity (Table 2).

Besides, the funnel plot was visually inspected for symmetry and there was an asymmetrical distribution of the effect estimates (Fig 3). To confirm this, egger's test was used and the test statistic showed the absence of significant publication bias ($p$ = 0.606).

## Subgroup analysis

Subgroup analysis by residence, sample size, and the cutoff point used to measure the outcome variable was carry out to identify the possible source variation between studies. The analysis showed that residence (both rural and urban) was one of the sources of severe heterogeneity ($I^2$ = 98.4, $P$ = 0.000) (Fig 4). However, severe heterogeneity still existed in the subgroup analysis of studies by sample size and the cutoff point used to measure the outcome variable.

The highest prevalence of short birth spacing (58.2%) was observed among studies conducted in rural settings (58.2%) as compared to both (41.1%) (Table 3).

**Table 2. Univariate meta-regression analysis to determine factors related to the heterogeneity of the prevalence of short birth spacing in Ethiopia, 2020.**

| Variables | Coefficient | P-value |
|---|---|---|
| Sample size | -0.0412 | 0.357 |
| Response rate | -1.3660 | 0.890 |
| Year of publication | -2.7533 | 0.546 |

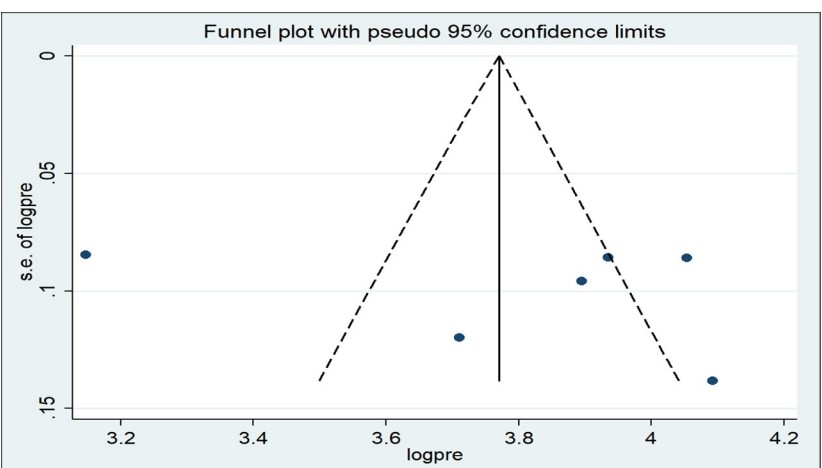

**Fig 3. Funnel plot with 95% confidence limits of the pooled prevalence of short birth spacing among reproductive-age women in Ethiopia, 2020.**

## Factors associated with short birth spacing

The association between contraceptive non-utilization and short birth spacing was determined based on the result of six studies [38, 39, 41–44]. The result indicated women who had never used contraception before the conception of the last child were 3.87 times more likely to have short birth spacing as compared to women who utilized contraceptive methods [OR = 3.87, 95%, CI: (2.29, 6.53)]. The random effect meta-analysis model was used to examine the association due to the presence of significant heterogeneity ($I^2$ = 90.4%, $p$<0.001) (Fig 5).

The effect estimates were distributed symmetrically on the funnel plot indicating the absence of publication bias (Fig 6). The result of Egger's tests also revealed the absence of publication bias ($p$-value = 0.684).

|  | Study |  | ES (95% CI) | % Weight |
|---|---|---|---|---|
| Authors | year |  |  |  |
| **Rural** |  |  |  |  |
| Ayane et al | 2017 |  | 59.87 (54.45, 65.29) | 16.43 |
| Yohannes et al | 2010 |  | 57.58 (54.18, 60.98) | 16.76 |
| Subtotal (I-squared = 0.0%, p = 0.483) |  |  | 58.23 (55.35, 61.11) | 33.18 |
| **Both** |  |  |  |  |
| Tsegaye et al. | 2016 |  | 51.17 (47.73, 54.61) | 16.75 |
| Shallo and Gobena | 2016 |  | 49.15 (45.30, 53.00) | 16.70 |
| Ejigu et al. | 2016 |  | 40.88 (36.12, 45.63) | 16.55 |
| Gebrehiwot et al | 2016 |  | 23.29 (20.36, 26.21) | 16.81 |
| Subtotal (I-squared = 98.4%, p = 0.000) |  |  | 41.10 (26.73, 55.47) | 66.82 |
| Overall (I-squared = 98.4%, p = 0.000) |  |  | 46.95 (34.75, 59.14) | 100.00 |

NOTE: Weights are from random effects analysis

0 50 100

**Fig 4. Forest plot of the pooled prevalence of short birth spacing by the area of residence (rural versus both urban and rural) in Ethiopia, 2020.**

**Table 3. Subgroup prevalence of short birth spacing in Ethiopia, 2020 (n = 6).**

| Variables | Characteristics | Included studies | Sample size | Prevalence (95% CI) |
|---|---|---|---|---|
| Residence | Rural and urban | 4 | 2,672 | 41.1 (26.7,55.5) |
| | Rural | 2 | 1,125 | 58.2 (55.3,61.1) |
| Sample size | ≤700 | 3 | 1,372 | 49.9 (40.1, 59.7) |
| | >700 | 3 | 2,425 | 44 (22.4,65.6) |
| Short birth spacing cut off point | <33 months | 3 | 1,861 | 37.7 (20.9,54.5) |
| | <36 months | 3 | 1,936 | 56 (50.8, 61.1) |

A total of three studies [39, 42, 43] were included to identify the association between the duration of breastfeeding and short birth spacing. The result of the random-effect meta-analysis showed the odds of practicing short birth spacing was 16.9 higher among women who breastfeed their child less than 24 months as compared to women who breastfeed ≥ 24 months [OR = 16.9, 95%CI: (2.69, 106.47)]. High heterogeneity was observed between the included studies ($I^2$ = 93.9%, p<0.001) (Fig 7).

The results of Egger's tests showed the absence of publication bias (*P* = 0.748) even though a slight asymmetrical distribution of the effect estimates was observed on the funnel plot (Fig 8).

The association between the educational status of the mother and short birth spacing was also assessed based on the results of three studies [39, 42, 43]. But, pooled estimate showed non-significant association [OR = 1.60, 95%CI: (0.59, 4.31)]. The included articles showed significant heterogeneity ($I^2$ = 94.9%, *p*<0.001) (Fig 9).

The presence of publication bias was assessed using both funnel plots and Egger's test. The effect estimates were distributed asymmetrically on the funnel plot indicating the presence of publication bias (Fig 10). But Egger's test objectively confirmed the absence of publication bias (*p* = 0.876).

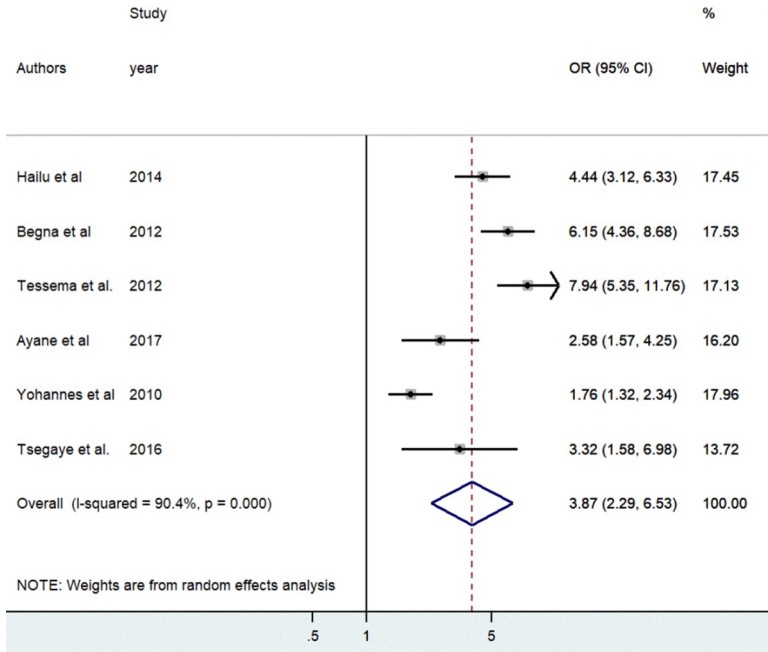

**Fig 5. The pooled odds ratio of the association between contraceptive use and short birth spacing in Ethiopia, 2020.**

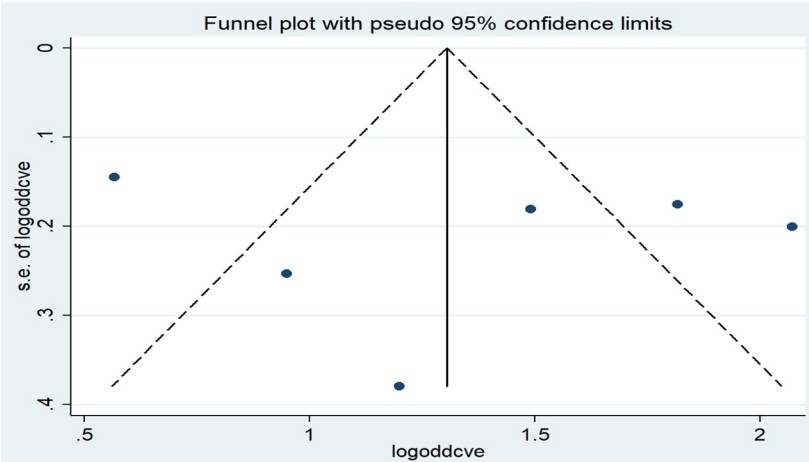

**Fig 6. Funnel plot with 95% confidence limits of the pooled odds ratio of contraceptive use among women in Ethiopia, 2020.**

## Discussion

Although WHO recommended optimal birth spacing to ensure the health of the mother and the newborn [8], 46.9%, [95% CI: (34.7, 59.1)] of reproductive age women practiced short birth spacing in Ethiopia. The finding of this study is in line with the Ethiopian Demographic and Health Survey (EDHS) report (41.5%) [14] and a study done in Tanzania (48.4%) [49]. But, the pooled prevalence of short birth spacing was higher than the studies conducted in Nepal (23%) [50], Bangladesh (24.6%) [18], and Iran (28.5%) [6]. This discrepancy could be due to variation in maternal socio-demographic characteristics, methodological variation

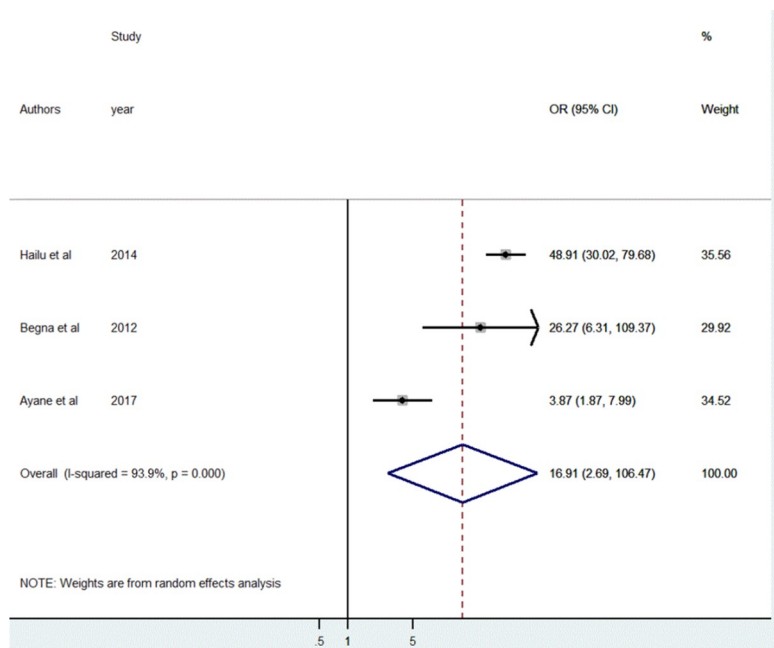

**Fig 7. The pooled odds ratio of the association between the duration of breastfeeding and short birth spacing in Ethiopia, 2020.**

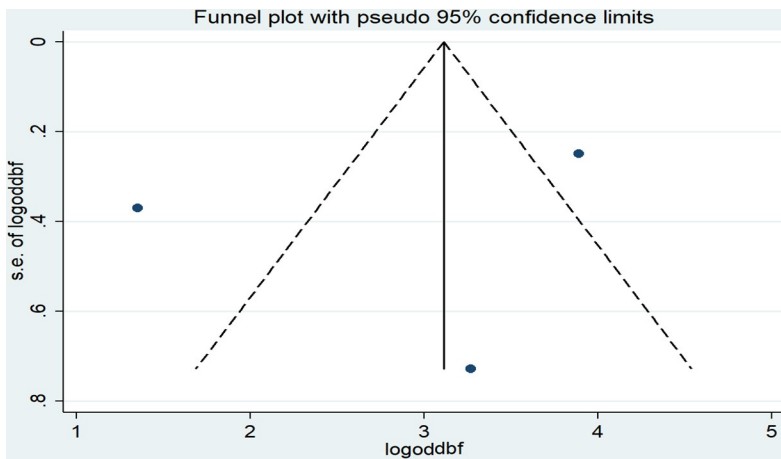

**Fig 8. Funnel plot with 95% confidence limits of the pooled odds ratio of the duration of breastfeeding among women in Ethiopia, 2020.**

(data analysis, study design, and sampling of study participants), the difference in economic status, lifestyle difference, and the difference in the health service utilization. Furthermore, varied study settings (urban and rural) and the difference in the methods applied to measure the outcome variable may have paramount importance for the variation of the findings.

Even if the increasing birth interval was the primary intervention area of the national government, practically, short birth spacing among reproductive-age women was still quite common. Babies delivered from those mothers who practice short birth spacing might experience low birth weight, small for gestational age, preterm birth, and congenital anomalies. Mothers are also at high risk of developing different adverse pregnancy outcomes [15–33]. So, the

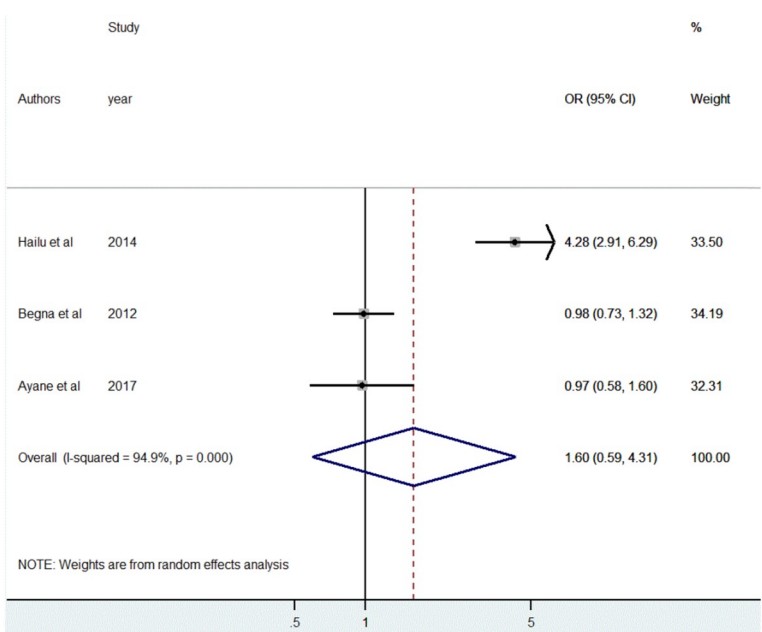

**Fig 9. The pooled odds ratio of the association between maternal educational status and short birth spacing in Ethiopia, 2020.**

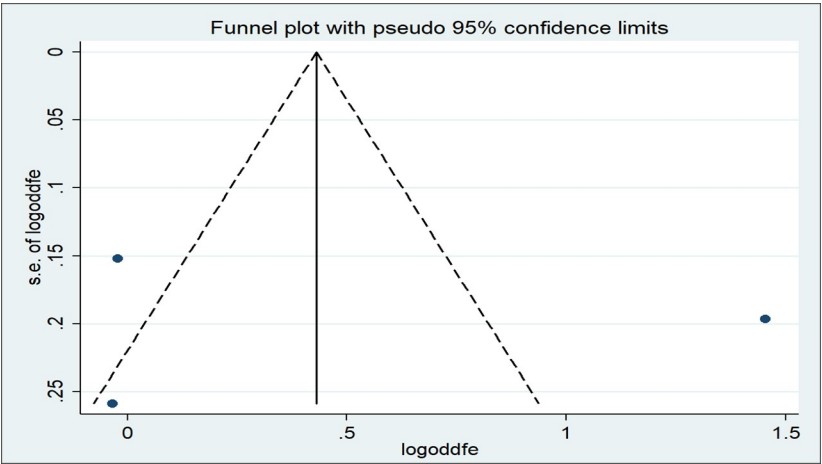

**Fig 10. Funnel plot with 95% confidence limits of the pooled odds ratio of maternal educational status in Ethiopia, 2020.**

clinician needs to be aware of different risk factors and should give special emphasis to them in addition to increasing family planning utilization and duration of breastfeeding among women of reproductive age women.

In this study, the result of subgroup analysis by residence (both urban and rural versus rural) indicated, the practice of short birth spacing was significantly higher in rural settings (58.2%) as compared to both (41.1%). This finding is in agreement with the recent EDHS report which indicated that rural women had a high prevalence of short birth spacing than urban women [4]. Access to the health facility, educational, and employment opportunities among women living in urban areas could be responsible for the variation by residence.

Contraceptive non-utilization had a significant association with short birth spacing. This is consistent with studies conducted in Jordan, Manipur, Egypt, Southeastern Nigeria, and Mbarara Hospital [51–55]. The objective of the family planning program is not only to limit the family size but also adequately space the children by delaying pregnancy. Whenever women use contraceptive methods, they are deliberately avoiding short birth spacing and unwanted pregnancy [9].

The duration of breastfeeding also had a significant association with short birth spacing. The finding is in line with studies conducted in Jordan and Southeastern Nigeria [51, 52]. During breastfeeding, a high amount of prolactin hormone is produced as a result of nipple stimulation. The prolactin hormone inhibits ovulation by reducing the release of the gonadotrophic hormone and resulted in post-partum amenorrhea and thereby lengthen the interval between births. On the other hand, a woman with a short duration of breastfeeding might experience short birth spacing as a result of a decreased level of prolactin concentration in the body [56].

## Limitations of the study

This systematic review and meta-analysis considered only articles written in English language. Besides, most of the articles included in the analysis were cross-sectional studies and had a small sample size, these might affect the pooled estimates. Conducting meta-analysis with low numbers of studies is another potential limitation of this study which decreases statistical power, permits large standard errors, and leads to publication bias. Moreover, regions may be under-represented since the included studies were from only four regions of Ethiopia.

## Conclusions

A significant number of Ethiopian women practiced short birth spacing though a minimum inter-pregnancy interval of two years was recommended by WHO. Duration of breastfeeding and non-use of contraceptives were factors affecting short birth spacing. Based on the finding, Health care providers should counsel women about the importance of optimal birth spacing, breastfeeding, and contraceptive utilization during their antenatal care, delivery, and postnatal care follow-up. Health extension workers should provide house to house education to improve contraceptive utilization and breastfeeding practice in the community.

## Supporting information

**S1 File. Search strategy used to estimate the pooled prevalence of short birth spacing and its association with contraceptive use, educational status, and duration of breastfeeding among reproductive-age women in Ethiopia.**
(DOCX)

**S1 Table. The Preferred Reporting Items for Systematic Reviews and Meta-Analysis (PRISMA) checklist.**
(DOC)

**S1 Dataset. The data set used to estimate the pooled prevalence of short birth spacing and its association with contraceptive use, educational status, and duration of breastfeeding among reproductive-age women in Ethiopia.**
(XLSX)

## Acknowledgments

The authors would like to acknowledge Wollo and Addis Ababa University Library for providing an available online database.

## Author Contributions

**Conceptualization:** Yitayish Damtie.

**Data curation:** Yitayish Damtie, Bereket Kefale, Melaku Yalew.

**Formal analysis:** Yitayish Damtie, Mastewal Arefaynie, Bezawit Adane.

**Investigation:** Yitayish Damtie, Melaku Yalew.

**Methodology:** Yitayish Damtie, Bereket Kefale, Melaku Yalew.

**Software:** Yitayish Damtie, Mastewal Arefaynie, Bezawit Adane.

**Supervision:** Bereket Kefale.

**Validation:** Yitayish Damtie, Bereket Kefale, Melaku Yalew.

**Writing – original draft:** Yitayish Damtie.

**Writing – review & editing:** Mastewal Arefaynie, Bezawit Adane.

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
