## [Decision Letter · Decision Letter 0]

2 Oct 2020

PONE-D-20-19678

Short birth spacing and its association with maternal educational status, contraceptive use, and duration of breastfeeding in Ethiopia. A systematic review and meta-analysis:

PLOS ONE

Dear Dr. Damtie,

Thank you for submitting your manuscript to PLOS ONE. After careful consideration, we feel that it has merit but does not fully meet PLOS ONE’s publication criteria as it currently stands. Therefore, we invite you to submit a revised version of the manuscript that addresses the points raised during the review process.

We look forward to receiving your revised manuscript.

Kind regards,

Mohammad Rifat Haider, MBBS, MHE, MPS, PhD

Academic Editor

PLOS ONE

Additional Editor Comments:

Take care of the reviewers' comments. Put special emphasis on the English language and grammar.

Journal Requirements:

2. Thank you for including the statement that "The database search was conducted on 'The searching of research articles was carried out from April 1 up to May 30, 2020, and articles published until May 30, 2020, were included in the review.' Please revise this statement to clarify whether all databases were searched from inception, or if there were any limits placed on the publication dates in your search.

3. At this time, we ask that you please provide the full search strategy and search terms for at least one database used as Supplementary Information.

4. Please provide the results of the assessment of publication bias conducted in the meta-analyses. Please provide the funnel plots of the Egger's regression test as a separate Figure in your manuscript.

Reviewers' comments:

Reviewer's Responses to Questions

**Comments to the Author**

1. Is the manuscript technically sound, and do the data support the conclusions?

Reviewer #1: Partly

Reviewer #2: Yes

2. Has the statistical analysis been performed appropriately and rigorously? 

Reviewer #1: Yes

Reviewer #2: Yes

3. Have the authors made all data underlying the findings in their manuscript fully available?

Reviewer #1: Yes

Reviewer #2: Yes

4. Is the manuscript presented in an intelligible fashion and written in standard English?

Reviewer #1: No

Reviewer #2: Yes

5. Review Comments to the Author

Reviewer #1: I thank the editors for the opportunity to review this manuscript. Short birth spacing is a major cause of morbidity among children and women from low- and middle-income countries, and it has a large impact in Sub-Saharan countries. Although the topic is highly relevant, major changes to the manuscript are required.

A significant limitation of the manuscript in its current form is the English. There are many mistakes and inconsistencies that require an urgent revision by a professional copyeditor. The mistakes make the manuscript challenging to read and understand.

Abstract

- in methods: “and determinants of birth spacing in Ethiopia...” is not true. The researchers prespecified the factors they explored. Please clarify.

- results: please describe the number of articles included in the review and in the meta-analyses.

- conclusion: "The practice of short birth spacing in Ethiopia was higher." Please elaborate on this phrase.

Introduction

- Given the considerable number of factors associated with short birth spacing, it is not clear why the authors chose to explore only maternal education, contraceptive use, and duration of breastfeeding. Please clarify what criteria were used to choose the aforementioned factors.

Methods

- Line 95: “Grey literature was identified through the input of content experts.” It is not clear who these experts were and how they were selected and/or approached. Moreover, it is not clear how many studies the experts included and what criteria was used to select the studies.

- Lines 99 to 105: the authors should include the exact search they used in the different databases. A list of terms is not enough. Moreover, please consider uploading the search as an additional file as this will save space.

- Line 105: “This systematic review and meta-analysis followed the Preferred Reporting Items for Systematic Reviews and Meta-Analysis (PRISMA) checklist.” The PRISMA checklist is preferred for systematic reviews and meta-analysis of experimental studies. The MOOSE guideline is preferred for Meta-Analysis of Observational Studies in Epidemiology (doi:10.1001/jama.283.15.2008). Please refer to the MOOSE reporting guidelines and use them to report the findings of your article.

- Inclusion criteria: the review is supposed to be focused only on studies exploring maternal education, contraceptive use, and duration of breastfeeding. However, this is not reflected in the inclusion criteria. Please clarify.

- Study setting: I suggest the authors to move this information to the introduction.

- Outcome measurement: There will be many different definitions or cut-off points among the three factors that the authors wanted to explore in their systematic review. For example, the authors will find studies exploring duration of breastfeeding with cut-off points of 6, 8, 12, 18, and 24 months. It is not clear why the authors pre-specified their cut-off point to be 24 months. Moreover, it is not clear why the authors pre-specified their education contrast to be formal education vs no formal education, or contraceptive use yes vs no. Pre-specifying these contrasts would exclude many articles. This should be explained in the limitations section. Additionally, this information is not explained in the inclusion criteria. Please clarify.

Results

- The authors reported high heterogeneity among the included studies. Why did the authors decide to conduct meta-analysis instead of providing a narrative synthesis of the results? Please clarify this sensitive issue.

- From the first paragraph of Results, it seems that the authors conducted a two-step screening phase. First, they screened the titles, and second, they screened the abstracts of the publications. In systematic reviews, however, titles and abstracts are evaluated together. I believe that screening the references in the way the authors did could introduce major biases in the review results. Please explain the rationale behind the two-step screening process.

- First paragraph of Results: the authors overexplain the flow diagram of the articles screened/included. This information could be easily summarized in Figure 1.

- In lines 181 to 184, the authors say that they included 7 studies in the meta-analysis. It is not clear why the authors included only these 7 studies from the original 9 included studies. More concerning, in lines 189 and 190 the authors mention that they included 6 articles, not 7. This issue needs clarification.

- The authors included only three studies in the meta-analysis related to breastfeeding. In the limitations, please explain the limitations of conducting meta-analysis with low numbers of studies.

Discussion

- Lines 228 & 229, and 247 & 248: there is no need to repeat the objective of the study.

- Line 241: “a subgroup analysis was done by residence (both urban and rural versus rural)” It is not clear to me why the authors decided to explore this contrast. Why not to simply explore urban vs rural? Please clarify.

- Lines 249 to 251 and 265 to 258: there is no need to repeat the results of the study.

Conclusion

- “Based on the finding, the Ethiopian government should improve the existing family planning program for all women of childbearing age.” Given the considerable limitations of this meta-analysis, I believe the authors are overstating their conclusion. Please adjust in light of the limitations of the study.

Reviewer #2: General comments: This is an important review aimed at estimating the pooled prevalence of short birth spacing and its association with maternal educational status, contraceptive use, and duration of breastfeeding in. However, a number of issues need attention before acceptable for publication.

ABSTRACT:

Objectives: This is Ok.

Methodology: The authors should include the key words used in the search.

The authors should state the selection criteria.

The authors should state the outcome measures assessed.

The authors should state the subjects of the studies they included in the meta-analysis.

The authors should state which outcomes were analysed.

Main results: The authors should state the number of studies assessed, identified and how many were included in the meta-analysis. What were the results of heterogeneity across studies?

CONCLUSION

Why did the authors state that: “The practice of short birth spacing in Ethiopia was higher.?’

The authors should rephrase their conclusion because the practice of short birth spacing in Ethiopia cannot he said to be higher.

The authors should avoid 'slangs' in their writing like ‘didn’t’.

INTRODUCTION:

Although, this is well written; however, the introduction needs brushing up of language and beefing up the justification for the study.

MATERIALS AND METHODS

Why did the authors exclude relevant articles with full texts of which unavailable after two email contacts of the corresponding authors?

RESULTS

In table 1, why was the prevalence of the last three publications not included?

DISCUSSION

The authors must correct the typo errors and other errors in the manuscript. The authors need English language editor.

The authors should discuss the clinical implications of their study findings.

The authors should also discuss the strengths of their study.

6. PLOS authors have the option to publish the peer review history of their article (what does this mean?). If published, this will include your full peer review and any attached files.

Reviewer #1: No

Reviewer #2: **Yes: **George Eleje

---

## [Author Response · Author response to Decision Letter 0]

30 Nov 2020

Mohammad Rifat Haider (MBBS, MHE, MPS, PhD)

RE: Manuscript ID: PONE-D-20-19678 (Short birth spacing and its association with maternal educational status, contraceptive use, and duration of breastfeeding in Ethiopia. A systematic review and meta-analysis).

Dear Dr. Mohammad,

Thank you very much for your email and the comments/suggestions of the reviewers and academic editor. We have looked at the comments and have revised our paper accordingly. We hope our paper improved as a result of incorporating reviewer’s and academic editor’s comments and suggestions.

Please find for your kind consideration the following:

A rebuttal letter that responds to each point raised by the academic editor and reviewer. The point by points responses are written in italic font style.

A revised manuscript with track changes.

A revised paper without tracked changes

The point by points responses are written in italic font style.

While hoping that these changes would meet with your favourable consideration, we are happy to hear if there are more comments and suggestions. Please do not hesitate to let us know if you have any questions.

Yours Sincerely,

Yitayish Damtie 

School of Public Health, Wollo University 

Dessie, Ethiopia

Tel:+251943517982

E-mail: yitutile@gmail.com

Point by point response

Additional Editor Comments

Take care of the reviewers' comments. Put special emphasis on the English language and grammar.

We have tried our best to improve it accordingly.

Journal Requirements:

We have tried to revise our manuscript according to PLOSONE formatting style.

2. Thank you for including the statement that "The database search was conducted on 'The searching of research articles was carried out from April 1 up to May 30, 2020, and articles published until May 30, 2020, were included in the review.' Please revise this statement to clarify whether all databases were searched from inception, or if there were any limits placed on the publication dates in your search.

Thank you for remembering an important issue. As it is stated above, database search was carried out from April 1 up to May 30, 2020, and articles published from the 1990 to May 30, 2020 were included in the review but we have missed to write it in the manuscript. 

3. At this time, we ask that you please provide the full search strategy and search terms for at least one database used as Supplementary Information.

We have tried to attach the full search strategy as supplementary information.

4. Please provide the results of the assessment of publication bias conducted in the meta-analyses. Please provide the funnel plots of the Egger's regression test as a separate Figure in your manuscript.

We have taken in to consideration and provide the funnel plots as a separate Figure in our 

Manuscript.

Reviewer Comments to the Author

Reviewer#1: 

A significant limitation of the manuscript in its current form is the English. There are many mistakes and inconsistencies that require an urgent revision by a professional copyeditor. The mistakes make the manuscript challenging to read and understand.

We have tried to improve it accordingly.

Abstract

methods: “and determinants of birth spacing in Ethiopia...” is not true. The researchers pre-specified the factors they explored. Please clarify.

We acknowledged the problem and tried to correct it accordingly in the abstract. 

Results: please describe the number of articles included in the review and in the meta-analyses.

We have tried to describe it accordingly in the main document.

Conclusion: "The practice of short birth spacing in Ethiopia was higher." Please elaborate on this phrase.

We have taken in to consideration and tried to improve it accordingly.

Introduction

Given the considerable number of factors associated with short birth spacing, it is not clear why the authors chose to explore only maternal education, contraceptive use, and duration of breastfeeding. Please clarify what criteria were used to choose the aforementioned factors.

Thank you for your comment. As we have tried to justify it in the introduction, a wide range of maternal and service-related factors that affect short birth spacing were identified from the included studies. But, we only examine the association between short birth spacing and maternal education, contraceptive use, and duration of breast feeding since they were the most frequently mentioned factors and have controversial finding among the included studies. In addition to this, these three factors are clinically significant factors affecting short birth spacing. As a result, we have interested to test their statistical significance and settle the controversial finding they have across the included articles.

Methods

Line 95: “Grey literature was identified through the input of content experts.” It is not clear who these experts were and how they were selected and/or approached. Moreover, it is not clear how many studies the experts included and what criteria was used to select the studies.

Thank you for your comment. Only one study with a case-control study design reporting determinants of birth spacing among women of reproductive age was identified from the input of the two content experts who are currently working in our organization and specialized in different areas of maternal and child health issues. They were selected based on their area of specialization and approached after explaining the objective of our study. The pre-set inclusion criteria’s were used to select the grey literature. 

Lines 99 to 105: the authors should include the exact search they used in the different databases. A list of terms is not enough. Moreover, please consider uploading the search as an additional file as this will save space.

We have tried to upload the exact search as an additional file accordingly.

 Line 105: “This systematic review and meta-analysis followed the Preferred Reporting Items for Systematic Reviews and Meta-Analysis (PRISMA) checklist.” The PRISMA checklist is preferred for systematic reviews and meta-analysis of experimental studies. The MOOSE guideline is preferred for Meta-Analysis of Observational Studies in Epidemiology (doi:10.1001/jama.283.15.2008). Please refer to the MOOSE reporting guidelines and use them to report the findings of your article.

The comment is accepted and addressed accordingly.

Inclusion criteria: the review is supposed to be focused only on studies exploring maternal education, contraceptive use, and duration of breastfeeding. However, this is not reflected in the inclusion criteria. Please clarify.

Yes, definitely. The review focuses only on studies exploring maternal education, contraceptive use, and duration of breastfeeding. However, explaining it as an inclusion criteria would exclude many articles exploring only the prevalence of short birth spacing but not the factors that we have mentioned above.

Study setting: I suggest the authors to move this information to the introduction.

The comment is accepted and addressed accordingly.

Outcome measurement: There will be many different definitions or cut-off points among the three factors that the authors wanted to explore in their systematic review. For example, the authors will find studies exploring duration of breastfeeding with cut-off points of 6, 8, 12, 18, and 24 months. It is not clear why the authors pre-specified their cut-off point to be 24 months. Moreover, it is not clear why the authors pre-specified their education contrast to be formal education vs no formal education, or contraceptive use yes vs no. Pre-specifying these contrasts would exclude many articles. This should be explained in the limitations section. Additionally, this information is not explained in the inclusion criteria. Please clarify.

Yes, different cut-off points were used to define or categorize the three factors. But in our case, the cut-off points are not pre specified from the inception. Rather they are determined to be 24 months for the duration of breastfeeding, formal vs no formal education for maternal educational status and contraceptive use vs not use for contraceptive utilization after the review process since the majority of the included articles used these cut-off points to define the these factors. There is no need to explain the cut of points as an inclusion criteria as it exclude many articles.

Results

The authors reported high heterogeneity among the included studies. Why did the authors decide to conduct meta-analysis instead of providing a narrative synthesis of the results? Please clarify this sensitive issue.

Thank you for raising an important issue. Yes it is undeniable that sever heterogeneity was reported among the included studies. The reason behind conducting meta-analysis instead of providing a narrative synthesis of the results in the presence of high heterogeneity is to achieve the ultimate aim of our study since the objective of our study is to determine the pooled prevalence of short birth spacing and its association with contraceptive use, educational status and duration of breastfeeding among reproductive age women. But, efforts like using random effect model, conducting subgroup analysis and meta-regression were considered to minimize the variation and to identify the source of heterogeneity among the included studies. 

From the first paragraph of Results, it seems that the authors conducted a two-step screening phase. First, they screened the titles, and second, they screened the abstracts of the publications. In systematic reviews, however, titles and abstracts are evaluated together. I believe that screening the references in the way the authors did could introduce major biases in the review results. Please explain the rationale behind the two-step screening process.

In the actual screening process, the title and the abstract of the articles were screened and evaluated together. You can confirm this by observing the PRISMA flow diagram (Fig1). But, to be specific, the number of articles excluded by the title and the abstract were narrated separately in the result section of the main document. We acknowledge the problem since it introduce wrong understanding for the readers and tried to revise it accordingly. 

First paragraph of Results: the authors over explain the flow diagram of the articles screened/included. This information could be easily summarized in Figure 1.

We have tried to minimize the narration accordingly.

In lines 181 to 184, the authors say that they included 7 studies in the meta-analysis. It is not clear why the authors included only these 7 studies from the original 9 included studies. More concerning, in lines 189 and 190 the authors mention that they included 6 articles, not 7. This issue needs clarification.

No, we have not said that we included 7 studies in the meta-analysis. Rather we said, a total of 9 studies (Shallo and Gobena, Ejigu et al, Tsegaye et al, Ayane et al, Gebrehiwot et al, Yohannes et al, Hailu et al, Begna et al and Tessema et al) were included in this meta-analysis. Among these studies, 6 of them (Shallo and Gobena, Ejigu et al, Tsegaye et al, Ayane et al, Gebrehiwot et al and Yohannes et al) were included to estimate the pooled prevalence of short birth spacing and 6 among the 9 studies (Tsegaye et al, Ayane et al, Yohannes et al, Hailu et al, Begna et al and Tessema et al) were included to exam the association between maternal education, contraceptive use, and duration of breast feeding and short birth spacing.

The authors included only three studies in the meta-analysis related to breastfeeding. In the limitations, please explain the limitations of conducting meta-analysis with low numbers of studies.

We acknowledge the problem and tried to explain the limitation in the main document.

Discussion

Lines 228 & 229, and 247 & 248: there is no need to repeat the objective of the study.

We have tried to correct it accordingly in the main document.

Line 241: “a subgroup analysis was done by residence (both urban and rural versus rural)”. It is not clear to me why the authors decided to explore this contrast. Why not to simply explore urban vs rural? Please clarify.

Thank you for your comment. This is based on the setting in which the studies were conducted. As we have stated it in the main document, six studies were included to estimate the pooled prevalence of short birth spacing. Among these studies, four of them considered both rural and urban women, two studies considered only rural women and no study considered only urban women. Due to this reason, subgroup analysis by residence was explored as both urban and rural vs rural.

Lines 249 to 251 and 265 to 258: there is no need to repeat the results of the study.

We have tried to avoid the repetition accordingly 

Conclusion

 “Based on the finding, the Ethiopian government should improve the existing family planning program for all women of childbearing age.” Given the considerable limitations of this meta-analysis, I believe the authors are overstating their conclusion. Please adjust in light of the limitations of the study.

We have tried to adjust it accordingly. 

Reviewer#2: 

ABSTRACT:

Objectives: This is ok.

Thank you

Methodology: The authors should include the key words used in the search, should state the selection criteria, outcome measures assessed, the subjects of the studies they included in the meta-analysis and which outcomes were analyzed.

Yes, all the things that you have mentioned above are very important and must be included in the methodology part. However, PLOSONE formatting style (300 words in the abstract section) limit us to incorporate all the aforementioned comments under the abstract section. As much as possible, we have tried our best to include some of them. 

Main results: The authors should state the number of studies assessed, identified and how many were included in the meta-analysis.

We have tried to address it accordingly. 

What were the results of heterogeneity across studies?

We have tried to include the result of heterogeneity as per your comment.

CONCLUSION: why did the authors state that: “the practice of short birth spacing in Ethiopia was higher.’ The authors should rephrase their conclusion because the practice of short birth spacing in Ethiopia cannot he said to be higher. The authors should avoid 'slangs' in their writing like ‘didn’t’.

We acknowledge the problem and we have taken in to consideration. 

INTRODUCTION

although, this is well written; however, the introduction needs brushing up of language and beefing up the justification for the study.

We have tried to improve it accordingly. 

MATERIALS AND METHODS

Why did the authors exclude relevant articles with full texts of which unavailable after two email contacts of the corresponding authors?

We exclude articles with full texts of which unavailable after two email contacts of the corresponding authors because we couldn’t find relevant variables on the abstract section to achieve the prime objective of our study like sample size and the number of women with short birth spacing. 

RESULTS

In table 1, why was the prevalence of the last three publications not included?

Yes, definitely. The prevalence of the last three publications was not included in the table. Because, prevalence of short birth spacing could not be calculated the in the case of studies with a case-control and retrospective follow up study design. But, we include these three studies to examine the association between short birth spacing with maternal educational status, contraceptive use, and duration of breastfeeding.

DISCUSSION

The authors must correct the type errors and other errors in the manuscript. The authors need English language editor.

We have tried to improve the type and grammatical errors accordingly.

The authors should discuss the clinical implications of their study findings.

We have accepted the comment and tried to discuss the clinical implication of our finding in the discussion section of the main document.

The authors should also discuss the strengths of their study.

Thank you for your comment. Discussing the strengths of this study is not as such important since our study shared all the strengths of the systematic review and meta-analysis.

---

## [Decision Letter · Decision Letter 1]

9 Dec 2020

PONE-D-20-19678R1

Short birth spacing and its association with maternal educational status, contraceptive use, and duration of breastfeeding in Ethiopia. A systematic review and meta-analysis:

PLOS ONE

Dear Dr. Damtie,

Thank you for submitting your manuscript to PLOS ONE. After careful consideration, we feel that it has merit but does not fully meet PLOS ONE’s publication criteria as it currently stands. Therefore, we invite you to submit a revised version of the manuscript that addresses the points raised during the review process.

Please take care of the reviewer's comment.

Please submit your revised manuscript by Dec 22, 2020. If you will need more time than this to complete your revisions, please reply to this message or contact the journal office at plosone@plos.org. Please include the following items when submitting your revised manuscript:

We look forward to receiving your revised manuscript.

Kind regards,

Mohammad Rifat Haider, MBBS, MHE, MPS, PhD

Academic Editor

PLOS ONE

Additional Editor Comments (if provided):

Please take care of the reviewer's comment.

Reviewers' comments:

Reviewer's Responses to Questions

**Comments to the Author**

1. If the authors have adequately addressed your comments raised in a previous round of review and you feel that this manuscript is now acceptable for publication, you may indicate that here to bypass the “Comments to the Author” section, enter your conflict of interest statement in the “Confidential to Editor” section, and submit your "Accept" recommendation.

Reviewer #1: All comments have been addressed

Reviewer #2: All comments have been addressed

2. Is the manuscript technically sound, and do the data support the conclusions?

Reviewer #1: Yes

Reviewer #2: Yes

3. Has the statistical analysis been performed appropriately and rigorously? 

Reviewer #1: Yes

Reviewer #2: Yes

4. Have the authors made all data underlying the findings in their manuscript fully available?

Reviewer #1: Yes

Reviewer #2: Yes

5. Is the manuscript presented in an intelligible fashion and written in standard English?

Reviewer #1: Yes

Reviewer #2: No

6. Review Comments to the Author

Reviewer #1: The authors have addressed all my comments. There are still a few English mistakes. For example, "Short birth spacing had been linked" (line 63) -> has been linked. "Data source and searching strategies" (line 107) -> search strategy. I suggest the authors use a professional editing service to fine tune the manuscript.

Reviewer #2: The authors have addressed the issues adequately. This is a systematic review aimed at estimating the pooled prevalence of short birth spacing and its association with contraceptive use, educational status, and duration of breastfeeding among

reproductive-age women in Ethiopia.

Title: This is OK.

Regarding the English language, the authors have endeavored to work on it.

7. PLOS authors have the option to publish the peer review history of their article (what does this mean?). If published, this will include your full peer review and any attached files.

Reviewer #1: **Yes: **Juan Pimentel

Reviewer #2: **Yes: **George Eleje

---

## [Author Response · Author response to Decision Letter 1]

22 Dec 2020

Mohammad Rifat Haider (MBBS, MHE, MPS, PhD)

RE-2: Manuscript ID: PONE-D-20-19678R1 (Short birth spacing and its association with maternal educational status, contraceptive use, and duration of breastfeeding in Ethiopia. A systematic review and meta-analysis).

Dear Dr. Mohammad,

Thank you very much for your email and the comments/suggestions of the reviewers and academic editor. We have looked at the comments and have revised our paper accordingly. We hope our paper improved as a result of incorporating reviewer’s and academic editor’s comments and suggestions.

Please find for your kind consideration the following:

A rebuttal letter that responds to each point raised by the academic editor and reviewer(s) (labeled as Response to Reviewers).

A marked-up copy of your manuscript that highlights changes made to the original version (labeled as Revised Manuscript with Track Changes).

An unmarked version of your revised paper without tracked changes (labeled as Manuscript).

The point by points responses are written in italic font style.

While hoping that these changes would meet with your favourable consideration, we are happy to hear if there are more comments and suggestions. Please do not hesitate to let us know if you have any questions.

Yours Sincerely,

Yitayish Damtie 

School of Public Health, Wollo University 

Dessie, Ethiopia

Tel:+251943517982

E-mail: yitutile@gmail.com

Point by point response

Additional Editor Comments

Please take care of the reviewer's comment.

Thank you for your comment and suggestion, we have tried to incorporate all the reviewers comment in our revised manuscript. 

Reviewer’s comments 

Reviewer #1: 

The authors have addressed all my comments. There are still a few English mistakes. For example, "Short birth spacing had been linked" (line 63) -> has been linked. "Data source and searching strategies" (line 107) -> search strategy. I suggest the authors use a professional editing service to fine tune the manuscript.

Thank you for your constructive comments. We acknowledge the problem and we have tried to change “Short birth spacing had been linked" to has been linked and Data source and searching strategies" to search strategy on line number 66 and 102 of our revised manuscript respectively. 

We have tried our best to address the English mistakes in our manuscript 

Reviewer #2: 

The authors have addressed the issues adequately. This is a systematic review aimed at estimating the pooled prevalence of short birth spacing and its association with contraceptive use educational status and duration of breastfeeding among reproductive-age women in Ethiopia. 

Title: This is ok 

Thank you in advance.

Regarding the English language, the authors have endeavored to work on it.

Thank you for your comment. Efforts were made to address all the English mistakes and grammatical errors in our revised manuscript.

---

## [Decision Letter · Decision Letter 2]

18 Jan 2021

Short birth spacing and its association with maternal educational status, contraceptive use, and duration of breastfeeding in Ethiopia. A systematic review and meta-analysis:

PONE-D-20-19678R2

Dear Dr. Damtie,

We’re pleased to inform you that your manuscript has been judged scientifically suitable for publication and will be formally accepted for publication once it meets all outstanding technical requirements.

Kind regards,

Mohammad Rifat Haider, MBBS, MHE, MPS, PhD

Academic Editor

PLOS ONE

Additional Editor Comments (optional):

Reviewers' comments:

Reviewer's Responses to Questions

**Comments to the Author**

1. If the authors have adequately addressed your comments raised in a previous round of review and you feel that this manuscript is now acceptable for publication, you may indicate that here to bypass the “Comments to the Author” section, enter your conflict of interest statement in the “Confidential to Editor” section, and submit your "Accept" recommendation.

Reviewer #2: All comments have been addressed

2. Is the manuscript technically sound, and do the data support the conclusions?

Reviewer #2: Yes

3. Has the statistical analysis been performed appropriately and rigorously? 

Reviewer #2: Yes

4. Have the authors made all data underlying the findings in their manuscript fully available?

Reviewer #2: Yes

5. Is the manuscript presented in an intelligible fashion and written in standard English?

Reviewer #2: Yes

6. Review Comments to the Author

Reviewer #2: The revised version has been well received. The corrections are clearly marked in track changes format. The authors have addressed the comments satisfactorily.

7. PLOS authors have the option to publish the peer review history of their article (what does this mean?). If published, this will include your full peer review and any attached files.

Reviewer #2: **Yes: **George Eleje

---

## [Editor Report · Acceptance letter]

21 Jan 2021

PONE-D-20-19678R2 

Short birth spacing and its association with maternal educational status, contraceptive use, and duration of breastfeeding in Ethiopia.  a systematic review and meta-analysis 

Dear Dr. Damtie:

I'm pleased to inform you that your manuscript has been deemed suitable for publication in PLOS ONE. Congratulations! Your manuscript is now with our production department. 

Kind regards, 

on behalf of

Dr. Mohammad Rifat Haider 

Academic Editor

PLOS ONE